# Extortion strategies resist disciplining when higher competitiveness is rewarded with extra gain

Lutz Becks [1,2] & Manfred Milinski [3]

Cooperative strategies are predicted for repeated social interactions. The recently described Zero Determinant (ZD) strategies enforce the partner's cooperation because the 'generous' ZD players help their cooperative partners while 'extortionate' ZD players exploit their partners' cooperation. Partners may accede to extortion because it pays them to do so, but the partner can sabotage his own and his extortioner's score by defecting to discipline the extortioner. Thus, extortion is predicted to turn into generous and disappear. Here, we show with human volunteers that an additional monetary incentive (bonus) paid to the finally competitively superior player maintains extortion. Unexpectedly, extortioners refused to become disciplined, thus forcing partners to accede. Occasional opposition reduced the extortioners' gain so that using extortion paid off only because of the bonus. With no bonus incentive, players used the generous ZD strategy. Our findings suggest that extortion strategies can prevail when higher competitiveness is rewarded with extra gain.

[1] Community Dynamics Group, Department of Evolutionary Ecology, Max-Planck-Institute for Evolutionary Biology, August-Thienemann-Strasse 2, 24306 Plön, Germany. [2] University of Konstanz, Mainaustraße 252, 78464 Konstanz, Germany. [3] Department of Evolutionary Ecology, Max-Planck-Institute for Evolutionary Biology, August-Thienemann-Strasse 2, 24306 Plön, Germany. Correspondence and requests for materials should be addressed to M.M. (email: milinski@evolbio.mpg.de)

Human cooperation is often based on reciprocity despite the risk of impending defection[1–3]. The paradigm for studying potential cooperation through reciprocation is the Prisoner's Dilemma (PD) game[1–3]. Each of two players can either cooperate (C) or defect (D). If both cooperate, each player earns more than if they both defect. However, if one defects and the other cooperates, the defector has the highest gain and the cooperator the lowest. Irrespective of what the other does you gain more by defection if the game is played only once, hence the dilemma. However, when the same subjects play the PD repeatedly, numerous sequences of, e.g., C, D, D, C, C and so on, are possible. Therefore the strategy that is most successful against any partner cannot be found by mathematical calculation because the array of potential strategies is too huge[4]. Axelrod's[3] computer tournament simulating evolution among strategies that had been proposed by theorists found 'Tit-for-Tat' as the winner (start with C, then copy your partner's previous move). The next champion was 'Generous Tit-for-Tat'[4], followed by 'Win-stay, lose-shift'[5], all largely cooperative strategies[6], though the daily newscasts report widespread uncooperative human behaviour.

Recently Press and Dyson[7] have dramatically changed our view on the Prisoner's Dilemma. They found a special class of strategies, called Zero Determinant (ZD) strategies, which enforce a linear relationship between the two players' scores. If the ZD player X chooses to extort Y, who is an 'evolutionary' player not possessing a theory of mind and instead simply seeks to adjust his strategy to maximise his own score in response to whatever X is doing, Y would not try to alter X's behaviour[8]. Extortionate strategies, reported by Press and Dyson[7], grant a disproportionate number of high payoffs to X at Y's expense. It is, however, in Y's best interest to cooperate with X because only by doing so Y is able to increase his own score. Thus, he ends up increasing X's score even more than his own. He will accede to X's extortion because it pays him to do so[8]. Extortioners use a conditional cooperative strategy with a bias to their own advantage (see Fig. 1 for an example). If, however, Y has a theory of mind, and sabotages both his own and X's score by defecting, he might hope to discipline X, as in an ultimatum game[9] with X proposing an unfair ultimatum and Y declining the offer thereby sabotaging the payoffs for both players[7]. In a usually one-shot ultimatum game (e.g. ref. [10]), disciplining has no future whereas in an iterated PD it may have. Theoretical and empirical studies thus predicted extortion turning into generous strategies[11–15].

Experimental studies have shown that human players accede to a computer playing extortion[16,17] for some time until they 'punish' the extortioner by playing D, sacrificing their small gain for tearing down the extortioner's larger gain, which could not coerce the computer. With evolutionary simulations theorists studied the new ZD world and predicted extortioners to switch to more cooperative, generous strategies. We thus expect again to see nice and cooperative strategies prevailing[8,11–15].

Extortioners risk being disciplined and have problems succeeding in evolving populations[11] because they end up with mutual defection when they meet each other. The more frequent they are the more likely it is to meet another extortioner. Thus there is a limit frequency under negative frequency-dependent selection beyond which more cooperative strategies have a higher overall payoff and can spread[11]. Stewart and Plotkin[11] have identified a different subset of ZD strategies, called 'generous ZD' strategies that forgive defecting opponents, but nevertheless dominate in evolving populations. An experimental confirmation of humans using generous ZD strategies is still elusive.

Contrary to extortionate behaviour a generous ZD player always starts with C, cooperates after mutual cooperation and only mildly punishes defection (see Fig. 1 for an example). Any deviation from mutual cooperation causes the generous player's payoff to decline more than that of her opponent. The regression of the co-player's payoff on the generous player's payoff yields a slope above the diagonal, thus generous players let their co-players succeed until both reach mutual cooperation. The payoff of the generous strategy never exceeds the payoff of the co-player. On the other hand, the regression of the co-player's payoff on an extortioner's payoff yields a slope below the diagonal, thus extortioners outcompete their co-players. The co-player's best response to both extortion and generous strategies is to cooperate.

Here we staged an experiment with students testing whether extortioners can be found and are disciplined, and whether 'generous ZD strategies' emerge thereafter. Additionally, we tested whether extortion can be favoured by offering one or both players an incentive to gain extra money if they manage to become competitively superior over their partner. Various kinds of incentives to try to gain more than an equal share exist in reality. Under completely symmetric power conditions we compared an iterated PD of 49 rounds (T0) with the same game either where an asymmetric incentive was assigned to one player chosen randomly (T1), but both players knew who had been chosen, or

| Decisions | | Final payoff |
|---|---|---|
| **a** Player acceding to extortion | C D C D C C D C C C C C C D C C C C C C C C C C C C C C C | 57 |
| **b** Extortionate strategy | D D D C D C C D C C D C C D D C D C C D C C D C C D C C D C | 102 |
| **c** Player not reactive to extortion | C D C D C D C D C D C D C D C D C D C D C D C D C D C D C D | 43 |
| **d** Extortionate strategy | D D D C D C D D D C D C D D D C D D D D C D D D C D D D D | 65 |
| **e** Player acceding to generous | C D C D C C D C C C C C C C C C C C C C C C C C C C C C C | 90 |
| **f** Generous strategy | C C D C C C C D C C C C C C C C C C C C C C C C C C C C C C | 85 |
| **g** Player not reactive to generous | C D C D C D C D C D C D C D C D C D C D C D C D C D C D C D | 84 |
| **h** Generous strategy | C C D C C C D C D C D C D C D C D C D C D C C C D C D C | 69 |

**Fig. 1** Reactive and non-reactive players playing either C or D, paired with extortionate or generous. **a** A reactive player paired with **b** an extortionate strategy; **c** a non-reactive strategy paired with **d** an extortionate strategy; **e** a reactive strategy paired with **f** a generous strategy; **g** a non-reactive strategy paired with **h** a generous strategy; extortionate strategies do not cooperate in the first round, they never cooperate after the partner's defection, they cooperate after the partner's cooperation with a probability of about 2/3. Generous strategies cooperate in the first round, they always cooperate after the partner's cooperation and they cooperate after the partner's defection with a probability of about 0.1, values according to (16). A player earns most against an extortionate strategy by giving in to extortion, providing the extortioner with a much higher payoff (**a**, **b**). Extortioners outcompete their co-players. A player earns most against a generous strategy by giving in to generous; generous players let their co-players succeed until both have a similar gain (**e**, **f**)

where the incentive was assigned to both players, but only one, the finally more competitive one, gained the extra bonus (T2). Our first hypothesis assumes that the incentive to earn an extra bonus is strong enough, especially when restricted to a designated player in T1, that this player adopts extortion to be competitively superior at the expense of some losses due to the partner's occasional sabotaging both his own and X's score by defecting. The extorting player may become disciplined after some time. In T2 any player could profit but only one could succeed. Extortion might be less expressed than in T1. Our second hypothesis is that in comparison to T1 and T2, T0 is completely egalitarian which might pave the way for generous ZD strategies to predominate because there is no bonus to gain that renders potential extortion especially profitable. We thus expect more cooperation in T0 and predict to see generous ZD strategies.

We find in egalitarian iterated Prisoner's Dilemmas with no incentive (T0) that generous ZD strategies prevail. When one player is assigned an additional monetary incentive (bonus) (T1), if he/she is the finally competitively superior player, he/she often adopts the extortion strategy. Unexpectedly, extortioners refuse to become disciplined by their partners, thus forcing partners to accede. When both players can reach the bonus (T2), the use of extortion is less pronounced.

## Results

**Cooperation, conflict, payoffs in treatments T0, T1, T2.** The proportion of cooperation, i.e. to play C over the 49 rounds, differed among treatments (GLM: $F_{2,48} = 4.9$, $p = 0.012$; mean ± s.e.m. per player: T0: $0.548 ± 0.21$; T1 = $0.35 ± 0.18$; T2 = $0.35 ± 0.17$) with significantly highest proportions in treatment T0 (post hoc test: T0−T1: $p = 0.02$; T0−T2: $p = 0.026$) and no difference between T1 and T2 (post hoc test: $p = 0.99$). Players with no incentive assigned (T0) were most cooperative. The payoff also differed significantly among treatments (GLM: $F_{2,48} = 5.78$, $p = 0.006$; mean ± s.e.m. per player and round: T0: €$0.231 ± 0.04$; T1: $0.189 ± 0.043$, T2: $0.19 ± 0.034$) with significantly higher payoffs in treatment T0 were players were assigned no incentive (post hoc test: T0−T1: $p = 0.007$; T0−T2: $p = 0.011$, T2−T1: $p = 0.99$). The occurrence of conflict, depicted by the proportion of DD decisions (i.e. both players decided D), differed among treatments (GLM: $F_{2,48} = 6.96$, $p = 0.0022$; mean ± s.e.m. per round: T0: $0.24 ± 0.15$; T1: $0.46 ± 0.24$, T2: $0.45 ± 0.18$) where T1 and T2 had higher proportions of DD compared to T0 (post hoc test: T0−T1: $p = 0.004$; T0−T2: $p = 0.006$, T2−T1: $p = 0.99$). Players with no incentive (T0) had the lowest rate of conflict. Thus, both the cooperation rate and the payoff were higher in T0 than in either T1 or T2 and the existence of an incentive, asymmetric or symmetric, to gain €10 extra when gaining 10% more than the partner over the 49 rounds decreased cooperation and income, and enhanced conflict.

Ten of 18 players (60%) in T1 assigned with the incentive of gaining €10 extra managed to earn 10% more than their co-players, whereas two players (10%) without that incentive earned 10% more than their co-players but did not receive €10 extra. Thus, among the 12 players who managed to earn 10% more than their co-player, more players had been assigned with the incentive (binomial test, two-tailed, $p = 0.039$). The incentive motivated competitive behaviour. Those assigned with the incentive made over all rounds of the game a higher proportion of D decisions ($0.693 ± 0.043$ per round) than their partners not assigned the incentive ($0.619 ± 0.05$ per round; Wilcoxon signed-rank matched pairs test, two-tailed: $z = −2.384$, $N = 18$, $p = 0.0171$).

**Test of 'generous ZD' strategy.** For deciding whether the generous strategy had been used in T0, we tested for linear

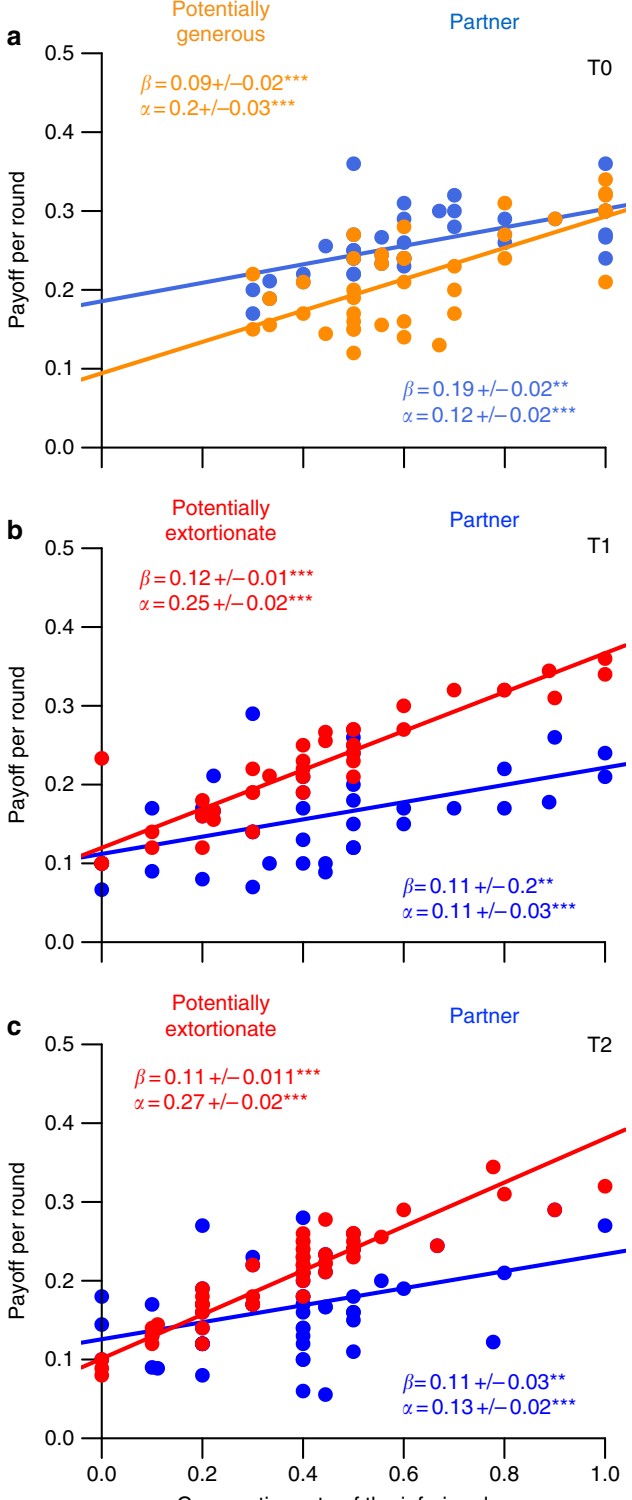

relationships between the payoffs of either player across many rounds of play and the cooperation rate of the potential generous player X; payoff ~ $\alpha \times$ cooperation rate of partner + $\beta$, where $\alpha$ is the slope and $\beta$ the intercept. For the 'generous' strategy to occur, the payoffs of both player X and partner Y must increase with X's cooperation. Furthermore, the payoff of the potential generous player X must increase more with her increasing cooperation compared to the increase of her partner Y's payoff. The payoffs of both players of a pair should not differ when cooperation is close

**Fig. 2** Payoffs of each player dependent on cooperation of inferior player. Correlation between cooperation rate of the inferior player and payoff per round for the inferior player and the other player in the treatments **a** without incentive (T0), the potentially generous player (orange) being inferior; **b** with asymmetric incentive (T1), the partner (blue) being inferior; and **c** with symmetric incentive (T2), the partner (blue) being inferior. Players have been selected as described in the Methods. Individual points represent average payoff and the average cooperation of 10 rounds per individual. Slope and intercept estimates (± s.e.m.) are derived from linear mixed effect models (see main text). **a** Inferior player potentially generous: $R^2 = 0.56$; partner $R^2 = 0.42$. **b** Inferior player partner of potentially extortionate: $R^2 = 0.85$; partner $R^2 = 0.37$. **c** Inferior player partner of potentially extortionate: $R^2 = 0.90$; partner $R^2 = 0.21$. For all other players, see Supplementary Figure 1

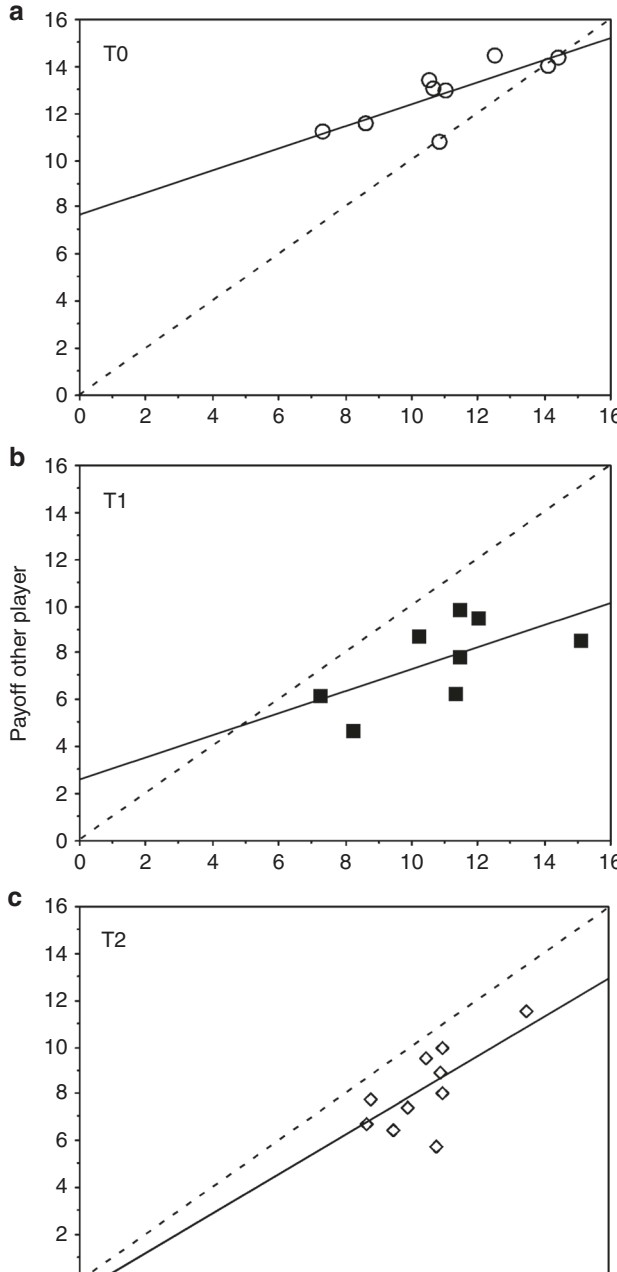

to mutual cooperation (intercept should be higher for partner Y than for generous player X).

Nine players fulfilled our criteria for 'generous' (see Methods). We found significant positive linear relationships between cooperation rate of the potential generous player X and payoff per round of either player (LME: payoff~cooperation rate; $F_{1,86} = 87.466$; $p = 9.33 \times 10^{-15}$; Fig. 2a, see also Supplementary Table 1) with a significantly lower increase for the partner of the potential generous player (LME: payoff~cooperation rate × player; $F_{1,86} = 6.418$; $p = 0.013$). Furthermore, the intercept was significantly lower for the potential generous player (LME: payoff~player; $F_{1,86} = 15.624$; $p = 0.00016$) as predicted for the generous strategy. Thus the generous players let their co-players succeed until they reached mutual cooperation and similar payoffs. The rest of the players in T0 do not seem to contain extortionate players (Supplementary Figure 1a, Supplementary Table 4).

**Test of 'extortion ZD' strategy.** For a proof of extortion in treatments T1 and T2, we need to show that with increasing cooperation of the partner Y the partner's own payoff increases and simultaneously that the payoff of the potential extortionate player X increases more. In that case, the partner Y can increase his payoff only by being more cooperative, thereby providing the extortionate player X with an increasingly higher payoff. At the same time, the payoffs should not be different when cooperation rates are close to zero (for criteria for 'extortionate player' see Methods).

For the treatment where one player per pair had an asymmetric incentive to earn the extra bonus (T1), we found significant positive linear relationships between cooperation rate of the partner and payoffs per round (LME: payoff~cooperation rate; $F_{1,176} = 120.42$; $p < 2.2 \times 10^{-16}$; Fig. 2b, see also Supplementary Table 2) with a significantly greater increase for the potential extortionate player (LME: payoff~cooperation rate × player; $F_{1,176} = 16.57$; $p = 0.00011$). Furthermore, there was no significant difference for the intercepts (LME: payoff~player; $F_{1,176} = 0.32$; $p = 0.57$). Thus, the player X who was assigned the incentive of gaining the extra bonus and gained it, used probably the extortion strategy enforcing cooperation of her partner Y who could increase his own gain only by increasing his cooperation supplying the extortioner with an increasingly higher gain. The rest of the players in T1 deviate from expectation for extortionate behaviour (Supplementary Figure 1b, Supplementary Table 5).

For the treatment where both players had an incentive to earn the extra bonus but only the player who would be 10% more competitive than the partner would receive it (T2), we found significant positive linear relationships between cooperation rate of the partner Y and payoffs per round (LME: payoff~cooperation rate; $F_{1,106} = 102.8$; $p < 2.2 \times 10^{-16}$; Fig. 2c, see also

**Fig. 3** Formal test of 'generous' and 'extortion'. Correlation between payoff of ZD player and the payoff of the partner in the treatments, **a** without incentive (T0), **b** with asymmetric incentive (T1), and **c** with symmetric incentive (T2); players have been selected as in Fig. 2; for statistics see main text

Supplementary Table 3) with a significantly greater increase for the potential extortioners X (LME: payoff~cooperation rate × player; $F_{1,106} = 14.12$; $p = 0.00028$). Furthermore, there was no significant difference for the intercepts (LME: payoff~player; $F_{1,176} = 0.26$; $p = 0.62$). Thus, the player X who gained the extra bonus probably used the extortion strategy enforcing cooperation of her partner Y who could increase his own gain only by increasing his cooperation supplying the extortioner X with an increasingly higher gain. There was no significant difference in the pay-off increase with increasing cooperation rate for the rest

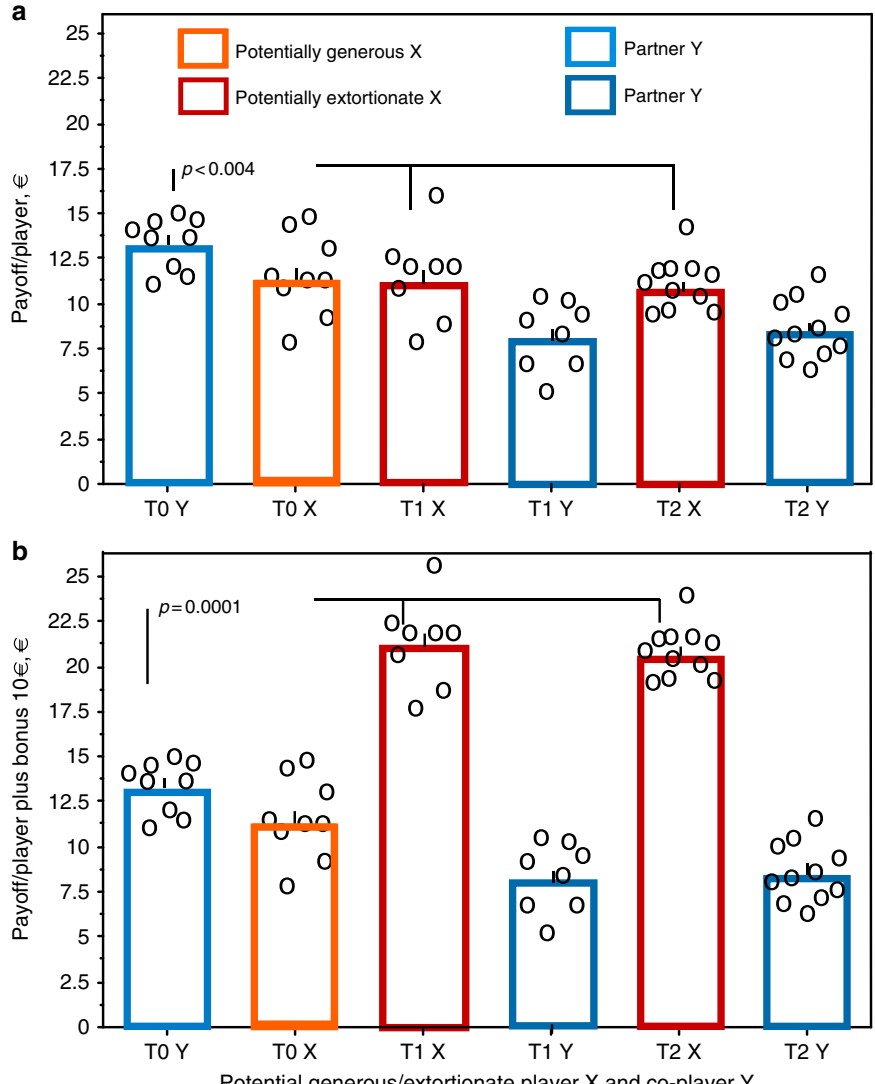

**Fig. 4** Payoff per player without and with bonus. Payoff (€) per player over 49 rounds in the treatment without incentive (T0), with asymmetric incentive (T1) and with symmetric incentive (T2). Players have been selected as in Fig. 2. Panel **a** shows payoffs without the additional bonus of 10€ added; *p* after two-tailed Mann–Whitney *U* test comparing the payoff of the partners of the potential generous ZD players in T0 with the payoff of the potential extortionate ZD players, pooled from the two treatments with incentive to earn the additional bonus T1 and T2; N1 = 9 (T0), N2 = 19 (T1, T2), Z = −2.902, *p* = 0.0037). Panel **b** shows payoffs with the additional bonus of 10€ added; Z = −4.208, *p* = 0.0001

of the players in T2; they thus deviate from expectation for extortionate behaviour (Supplementary Figure 1c, Supplementary Table 6).

**Comparison of payoffs of the potential ZD strategies**. We further compared the payoffs of the potential ZD strategy X against that of the partner Y (Fig. 3) to test for generous and extortionate strategy in the different treatments. For a proof of a generous ZD strategy the regression of the co-player's payoff on the generous player's payoff must yield a slope above the diagonal[16], thus generous players let their co-players succeed until both reach mutual cooperation. In T0 the symbols are above the diagonal (Fig. 3a), i.e. potential generous players let their partners succeed and the intercept of the regression line is far above zero (GLM with family Gamma: slope: $t = -2.629$, df = 7, $p = 0.034$; intercept: $t = 9.806$, df = 8, $p = 2.43 \times 10^{-5}$), which is required for a generous ZD strategy. For a proof of the extortionate strategy, the regression of the co-player's payoff on an extortioner's payoff must yield a slope below the diagonal[16], thus extortioners

outcompete their co-players. In T1 (Fig. 3b) and T2 (Fig. 3c) the symbols are below the diagonal, i.e. potential extortionate players outcompeted their partners and the intercept of the regression line is close to zero, which is in approximate agreement with requirements for an extortionate ZD strategy (T1: GLM with family Gamma: slope: $t = 1.98$, df = 6, $p = 0.0948$; intercept: $t = 0.982$, df = 7, $p = 0.3640$, Fig. 2b; T2: slope: $t = -3.02$, df = 8, $p = 0.0166$; intercept: $t = 5.7$, df = 9, $p = 0.0005$, Fig. 3c). As the intercept in T2 is only close to zero, we cannot prove strict extortion formally here.

**Extortioners refuse to be disciplined**. Extortion-like players in T1 and T2 earned, without the bonus added, significantly less than generous-like players in T0 during the PD game (Fig. 4a), even though they had outcompeted their co-players. Only when the extra bonus of €10 is added to their gain, it is obvious that being extortionate paid off (Fig. 4b). Even though the co-players of extortioners could have gained more by being fully cooperative, they opposed to being exploited responding often with D thus

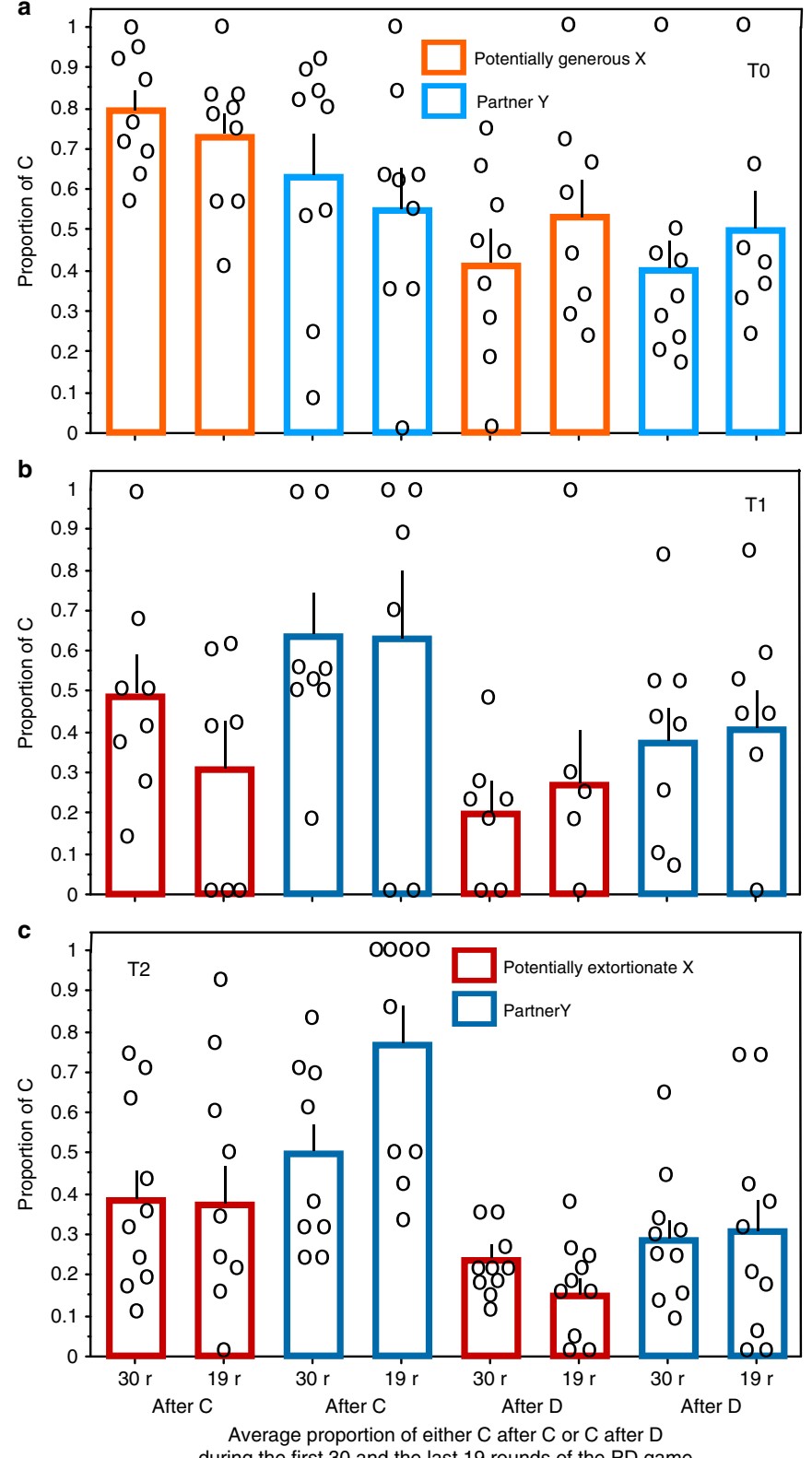

**Fig. 5** Change of cooperation during the game. Average proportion (+s.e.m.) of C (=cooperation) decisions after either co-player's C or D (=defect) during the first 30 and the last 19 rounds of the PD game. **a** T0—iterated Prisoner's Dilemma; C decisions of potentially generous ZD players X and their partners Y. **b** T1—iterated Prisoners Dilemma with asymmetric incentive to earn 10€ bonus, if pre-determined players achieves to earn at least 10% more than partner during the whole game. C decisions of potentially extortionate ZD players X and their partners Y. **c** T2—as T1 but any player of each Prisoner's Dilemma pair can earn the extra bonus dependent of who achieves a competitive advantage to earn at least 10% more than partner during the whole game

paying for their 'costly punishment' (cf. 16). Extortioners refused, however, to become disciplined. Their proportion of C after partner's C would increase during the game in that case. They did, however, not increase to respond with C after their partners' C from the first 30 to the last 19 rounds of the PD game, instead in all but one group extortioners decreased cooperation (e.g., in T1: $p < 0.005$, Fisher's Exact Test, Fig. 5b first two columns). In T2 they did not increase C after their partners' C from the first 30 to the last 19 rounds of the PD game either, but they did not decrease it (Fig. 5c) as in T1. Extortioners thus refused to become disciplined; they did not increase their cooperation in the last 19 rounds of the game.

**Reputation of generous and extortionate players**. A good reputation is worth having under many conditions[18]. Do extortioners damage their reputation because they acted as extortioners? Do generous players improve their reputation? After the experiment all players were asked on a questionnaire 'would you play again with your partner?' Answers were given under their pseudonym on a 7 point scale with 1 = 'I would very much like to' to 7 = 'not at all'. Answers from all pairs of players included in Fig. 1 (and Fig. 2) were analysed. In T0 generous players received $2.0 \pm 0.333$ (mean ± s.e.m.) points from their partners, their partners were allocated $4.44 \pm 0.71$ points (Wilcoxon matched pairs signed-rank test, two-tailed: $z = -2.375$, $p = 0.018$, $N = 9$). Thus, generous players gained a positive reputation, whereas they ranked their partners behaviour as almost neutral. In T1 extortioners received a negative score ($5.125 \pm 0.766$ points). However, they rated their partners whom they had enforced to be cooperative as positive ($2.75 \pm 0.62$ points; $z = -2.136$, $p = 0.033$, $N = 8$). In T2, where both players had the incentive to compete for the extra bonus, extortioners received almost the same almost neutral rating ($3.455 \pm 0.493$ points) as their competing partners ($3.818 \pm 0.63$ points, $z = -0.463$, $p = 0.644$, $N = 10$). As both players tried to be more successful and one of them managed to dominate the other, the similar score reflects the tension between them.

## Discussion

An evolutionary player trying to maximise his own gain would grant a disproportionate number of high payoffs to an extortionate co-player[7]. However, 'if the player has a 'theory of mind' about his co-player X, his only alternative to accepting positive, but meagre, rewards is to refuse them, hurting both himself and X'. 'He does this in the hope that X will eventually reduce her extortion factor', as Press and Dyson[7] suggested. Accordingly, evolutionary simulations predicted extortioners to switch to more cooperative strategies[8,11–15]. Extortion would no longer be expected. We confirm this prediction for the ordinary public goods game (T0). However, when one player could gain an extra bonus by being competitively superior (T1), she obviously used extortionate behaviour, which was opposed by the co-player with defection, probably trying to coerce the extortioner. It is well known that a fraction of the human population cares for equitable outcomes and exhibits a strong aversion against disadvantageous inequity[19], i.e., a response known as 'inequity aversion'[20]. Unexpectedly, however, X refused to be coerced and was steadfast extortionate, thus forcing her co-player to accede (Fig. 5b, c). Even though both players lost money, extortion paid off because of an extra bonus ahead (Fig. 4b).

While Fig. 2 clearly shows that extortion-like behaviour is occurring in the two asymmetric conditions and generous-like behaviour is occurring in the control condition, nonetheless the extortion-like behaviour is still not strictly extortion in the originally defined sense of Press and Dyson[7] and subsequent theoretical studies. This suggests that extortion and generosity appear

as spectrum in this study and indeed there are even cases particularly in T2, where low-levels of cooperation lead to the extortion-like strategy to score less then their partner. In this sense players may be thought of as permitting 'regions' of generosity and regions of extortion in their interactions, with the out-of game pay-off shifting the balance of what players are willing to accept.

Taken together, our results suggest that the co-players' opposition to being exploited was sufficient to reduce the extortionate players' potential gain to a level that is not worth using extortion without the expectation of an extra bonus. They would have gained more being generous players, as is shown in the control treatment (T0) with no promised incentive. When there was extra money to gain (T1, T2), extortioners unexpectedly resisted becoming disciplined. Thus extortioners with incentive were steadfast and reached the bonus despite losing money due to their co-players 'occasional' punishment'. Finally they earned much more with the bonus added than did generous players in the control treatment. The message of the present study seems to be that extortion strategies may be expected, when there is an incentive to gain an extra bonus through being more competitive. To resist being disciplined paid off through the extra money they actually gained. In reality, many opportunities exist to gain from being competitively superior, such as for collaborating colleagues in companies to become promoted to a rare better-paid higher position. Our competitive societies may favour extortion. Previous studies found extortion when either a power asymmetry between partners existed that prevented opposition[21] or players would lose out completely by not acceding to extortion by elected representatives[22]. The generous strategy is to be expected whenever some opposition is enough to reduce the extortioner's payoff so that it does not pay to be extortionate. Even though it pays the partner to accede to extortion, it would pay him more if the extortioner becomes 'generous'. In egalitarian iterated Prisoner's Dilemmas we expect generous ZD strategies to prevail as in the present study in the control treatment with no incentive. Our findings elucidate the relevance of extortion strategies to human behaviour and the role of incentive structures in inducing such behaviours.

## Methods

**Experimental design**. Experiments were conducted in November 2016 with 102 first-year biology students at the University of Kiel, Germany. All subjects gave their informed consent to participate. We invited 6 volunteers to each of 17 experimental sessions (by running three experimental games per session in parallel we aimed to create a more anonymous environment, with only two players anonymity is not possible). Before each session, subjects were orally informed about how to operate the computers, and about the measures that were taken to ensure the subjects' anonymity. During the experiment, subjects had a pseudonym, were separated by opaque partitions and they were instructed not to talk to each other during or after the experiment. Subjects made all their decisions through a computer interface based on z-Tree[23]. Each desk was equipped with paper and pencil, and subjects were encouraged to take notes. Subjects knew they would receive their earnings anonymously under their pseudonym and in cash directly after the game. Sessions took approximately 90 min including the initial instruction phase. For a detailed description of each treatment, a translation of the instructions is provided in the Supplementary Information.

**Treatments**. Treatment T0 is an Iterated Prisoner's Dilemma game with 49 rounds. In addition, in T1 one randomly chosen player was assigned the incentive to earn extra money, in T2 both players were assigned the incentive but only the competitively superior one gained the extra money. The three treatments have in common that each player interacts with only one partner in an iterated PD over 49 rounds (due to a software problem we had to exclude the 50th round from the analysis). To avoid end-round effects, participants were not informed about the number of rounds to be played. The three PD groups of two players each of an experimental session played the same treatment. The three PD games of a session were synchronised: after all six players had made their decisions in each round, the subjects of each pair were informed about their partner's decision, their own and the partner's payoff. Our statistical unit is the two players forming the prisoner's dilemma group since they could not interact with the other participants of the

experimental session in any way. In each round of the iterated PD the two partners had to simultaneously decide whether to cooperate (C) or to defect (D). If both cooperated, they each received a reward (R = €0.30). If one defected and the other cooperated, the defector got the temptation payoff (T = €0.50) and the cooperator obtained the sucker payoff (S = €0.00). However, if both defected they each received a low payoff (P = €0.10).[3] In all treatments each player of the PD group would receive a potential bonus payment of €5.00 if the pair had earned more than €0.30 per round throughout the experiment to reduce the risk of the pair to get stuck in DD, which is not informative.

The control treatment (T0) consisted of 15 PD groups in five experimental sessions with 49 rounds of iterative PD. The asymmetric treatment (T1) consisted of 18 PD groups in six experimental sessions with 49 rounds of iterative PD. One of the two players was randomly assigned the incentive to be able to gain an extra bonus of €10 if he/she had earned at least 10% more compared to the partner after all PD rounds. Both players were informed about who could win the extra bonus. The symmetric treatment (T2) consisted of 18 PD groups in six experimental sessions with 49 rounds of iterative PD. Again one player could earn the extra bonus of €10 if he/she had earned at least 10% more compared to the partner after all PD rounds. Different from the asymmetric treatment, where only the randomly chosen player could earn the extra bonus, both players in T2 had the potential to earn the extra bonus, but they knew that only one could achieve it.

**Test for generous ZD in T0 and extortionate ZD in T1, T2.** Extortion strategies belong to a class of iterated game strategies known as ZD, under which the probability of a player cooperating in a given round of the iterative PD is a function of the payoffs received by a player and her opponent. The result of a player X using such a strategy against an opponent Y is to induce a linear relationship between her expected payoff across many rounds of play (payoff X) and that of her opponent, (payoff Y), payoff Y = α payoff X + β, where α is the slope and β is the intercept. A strict 'extortion' strategy arises when β = 0 and 0 < α < 1 because in this case payoff X > payoff Y, and X extorts Y by forcing her to accept a lower payoff than X, or else receive 0 payoff. Note, however, that if one chooses β = (1 − α) R, where R is the payoff for mutual cooperation in a given round of the IPD, that the relationship between payoffs is inverted with payoff X < payoff Y. This is a 'generous' strategy under which a player X does worse than her partner Y if they deviate from mutual cooperation.

Thus, the generous strategy is observed when

$\alpha_{generous} > 0$,
$\alpha_{partner} > 0$,
$\alpha_{generous} > \alpha_{partner} > 0$
and $\beta_{generous} < \beta_{partner}$.
The extortion strategy is observed when

$\alpha_{extortion} > 0$,
$\alpha_{partner} > 0$,
$\alpha_{extortion} > \alpha_{partner} > 0$
and $\beta_{extortion} = \beta_{partner}$.

**Data analyses.** We used linear mixed effect models (LME) to test for potential generous in T0 and potential extortionate players in T1 and T2 using treatment-specific models (Fig. 2). Specifically, we correlated payoff per round and cooperation rate of the player (T0) or partner (T1, T2) with the other player as covariate. We included only those pairs in this statistical test that were selected according to the following rules. We tested whether players had used the generous strategy, who had started with C, had earned less than the partner, and had responded to partner Y's C in most cases (>0.5) with C (proportion C after C = 0.751 ± 0.05 (mean ± s.e.m.), range 0.565–0.979). Their behaviour is shown in Fig. 2a while the behaviour of the unselected players is shown in the Supplementary Figure 1a. We tested whether the player had used the extortion strategy, who managed to earn at least 10% more than the partner and who's partner had not earned less in the block in which he had cooperated most (the latter would not occur when playing against an extortionate player). Their behaviour is shown in Fig. 2b, c and the behaviour of the unselected players is shown in Supplementary Figure 1b, c.

For this statistical analysis we used the average payoff and the average cooperation of the 10 rounds of each period (9 rounds for the last period). So for each individual we had five data points for the payoff and the cooperation, respectively. We ran treatment-specific tests where we added pair (prisoners dilemma pair) nested within group (experimental group) as random effect to all models (e.g., payoff ~ cooperation rate partner × players + (1| group/pair). Statistical tests were performed in R (R Core Team[24]) using the lmerTest package[25]. $R^2$ were calculated using the r.squaredGLMM function of the MuMIn package[26].

We used treatment-specific generalised linear models (GLMs) with family Gamma (link = 'log') for the correlations between payoffs of the ZD player and the other player (Fig. 3). We used the Gamma family as data were non-normal distributed (shaprio.test: W = 0.98688, p = 0.0002) and positively bound. We used also GLMs with family Gamma (link = 'log') for the comparisons of the probability to cooperate, the proportion of DD decisions (i.e. both players decided D), and

payoff per round and player between treatments. To determine differences between the individual treatments, we used posthoc tests with corrections for multiple comparisons with the multcomp package[27].

**Ethics.** All experimental procedures follow German regulations. For behavioural experiments as ours, neither approval from an ethics committee nor written consent to participate is required according to German regulations. However, we have received the subjects' written informed consent to participate; full anonymity and true information was guaranteed, and subjects had the option to stop participating at any time.

**Reporting Summary.** Further information on experimental design is available in the Nature Research Reporting Summary linked to this Article.

## Data availability

Data are available from the Dryad Digital Repository: https://doi.org/10.5061/dryad.dq6cv73.

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

## Acknowledgements

We thank the students from the university of Kiel for their participation, H. Brendelberger and D. Semmann for logistic support, D. Semmann for the Z-tree program and help with performing the experiment, C. Hilbe and A. Sanchez for discussion. This study was supported by the Max Planck Society for the Advancement of Science.

## Author contributions

M.M. conceived the study, designed and performed the research, M.M. analysed data, L. B. performed statistics and calculated LME, M.M. wrote the paper, and L.B. and M.M. revised the manuscript and gave final approval for publication.

## Additional information

**Competing interests:** The authors declare no competing interests.

