## [Peer Review File · Nature Communications]

Reviewers' comments:

Reviewer #1 (Remarks to the Author):

What are the major claims of the paper?

This manuscript reports an experiment using the iterated prisoner's dilemma with or without incentives to compete (i.e., to obtain a payoff larger than the payoff of the other person). The main result is that incentives to compete favour the emergence of extortionate strategies. In other words, while in the baseline, extortioners can be disciplined by inequity averse players that renounce to part of their payoff to reduce payoff inequalities, this does not happen if incentives to compete are high enough.

Are they novel and will they be of interest to others in the community and the wider field?

To the best of my knowledge, this work is novel. Moreover, it would certainly be of interest to others in the community and the wider field.

Is the work convincing, and if not, what further evidence would be required to strengthen the conclusions?

Yes, the work is convincing.

On a more subjective note, do you feel that the paper will influence thinking in the field?

This is probably the weak part of the manuscript. To be honest, I was not terribly surprised by the results. There is large evidence that some people are inequity averse. Thus I find it quite obvious that these people can discipline extortioners, but only when extortioners do not have high incentive to compete. Thus, I am not sure whether this article is appropriate for a high impact journal publishing groundbreaking research, such as Nature Communications.

Appropriateness of statistical analysis

Yes

Reproducibility

I could not find exact experimental instructions and exact experimental procedure (e.g., time of day in which the experiment was conducted). So, as it stands now, the experiment cannot be replicated. However, I think this is pretty easy to fix.

Other comments:

The paper is generally well written, but I think it lacks some theoretical background: as I mentioned before, there is evidence that some people are inequity averse (Fehr & Schmidt, 1999). The existence of these people can certainly explain the results quite straightforwardly. I think that adding a discussion around would improve the paper. Reviewers' comments:

Reviewer #1 (Remarks to the Author):

What are the major claims of the paper?

This manuscript reports an experiment using the iterated prisoner's dilemma with or without incentives to compete (i.e., to obtain a payoff larger than the payoff of the other person). The main result is that incentives to compete favour the emergence of extortionate strategies. In other words, while in the baseline, extortioners can be disciplined by inequity averse players that renounce to part of their payoff to reduce payoff inequalities, this does not happen if incentives to compete are high enough.

Are they novel and will they be of interest to others in the community and the wider field?

To the best of my knowledge, this work is novel. Moreover, it would certainly be of interest to others in the community and the wider field.

Is the work convincing, and if not, what further evidence would be required to strengthen the conclusions?

Yes, the work is convincing.

On a more subjective note, do you feel that the paper will influence thinking in the field?

This is probably the weak part of the manuscript. To be honest, I was not terribly surprised by the results. There is large evidence that some people are inequity averse. Thus I find it quite obvious that these people can discipline extortioners, but only when extortioners do not have high incentive to compete. Thus, I am not sure whether this article is appropriate for a high impact journal publishing groundbreaking research, such as Nature Communications.

Appropriateness of statistical analysis

Yes

Reproducibility

I could not find exact experimental instructions and exact experimental procedure (e.g., time of day in which the experiment was conducted). So, as it stands now, the experiment cannot be replicated. However, I think this is pretty easy to fix.

Other comments:

The paper is generally well written, but I think it lacks some theoretical background: as I mentioned before, there is evidence that some people are inequity averse (Fehr & Schmidt, 1999). The existence of these people can certainly explain the results quite straightforwardly. I think that adding a discussion around would improve the paper.

Reviewer #2 (Remarks to the Author):

“Incentive for extra gain helps “extortion” to resist disciplining, without incentive “generous” wins” by Becks and Milinski explores the impact of asymmetry in incentives in an iterated Prisoner’s Dilemma (IPD) game on the strategic behavior of human players. They show that such asymmetry results in lower rates of cooperation, more competitive behavior, lower payoffs during the game and a tendency among the most competitive players to employ “extortion” strategies. In contrast they show that players tend to default to “generous” strategies when such asymmetry is absent. This is a fascinating study whose results are of both theoretical and practical interest since they elucidate the relevance of extortion strategies to human behavior and the role of incentive structures in inducing such behaviors. I therefore believe that this study will be of considerable interest to the broad readership of Nature Communications and I support publication.

Major comments

The authors are careful to discuss their strategies as “potentially generous” and “potentially extortionate” and this is correct since they are projecting the player’s actual behavior onto the space of ZD strategies. One striking aspect of their work is that, while figure 2 clearly shows that extortion-like behavior is occurring in the two asymmetric conditions and generous-like behavior is occurring in the control condition, nonetheless the extortion-like behavior is still not strictly extortion in the originally defined sense of Press & Dyson 2012 and subsequent theoretical studies. This is not to dispute their findings but simply to point out that extortion and generosity appear as spectrum in their work and indeed there are even case particularly in T2, where low-levels of cooperation lead to the extortion-like strategy to score less than their partner. In This sense players may be thought of as permitting “regions” of generosity and regions of extortion in their interactions, with the out-of game pay-off shifting the balance of what players are willing to accept. I think this aspect of the empirical results is worth highlighting as it is interesting and should inform the way subsequent theoretical, computational and experimental studies are structured.

We thank the reviewers for their positive and constructive comments, which we were delighted to follow for improving the manuscript.

Reviewers' comments:

Reviewer #1 (Remarks to the Author):

What are the major claims of the paper?

This manuscript reports an experiment using the iterated prisoner's dilemma with or without incentives to compete (i.e., to obtain a payoff larger than the payoff of the other person). The main result is that incentives to compete favour the emergence of extortionate strategies. In other words, while in the baseline, extortioners can be disciplined by inequity averse players that renounce to part of their payoff to reduce payoff inequalities, this does not happen if incentives to compete are high enough.

Are they novel and will they be of interest to others in the community and the wider field?

To the best of my knowledge, this work is novel. Moreover, it would certainly be of interest to others in the community and the wider field.

Is the work convincing, and if not, what further evidence would be required to strengthen the conclusions?

Yes, the work is convincing.

On a more subjective note, do you feel that the paper will influence thinking in the field?

This is probably the weak part of the manuscript. To be honest, I was not terribly surprised by the results. There is large evidence that some people are inequity averse. Thus I find it quite obvious that these people can discipline extortioners, but only when extortioners do not have high incentive to compete. Thus, I am not sure whether this article is appropriate for a high impact journal publishing groundbreaking research, such as Nature Communications.

We completely agree with the conclusion that opposing to extortion is a case of inequity aversion cf Fehr & Schmidt 1999. Here in our repeated game it has a purpose, namely to potentially discipline extortioners. We agree that this finding is not unexpected, actually predicted by Dyson and Press (they do not call it inequity aversion). The expectation of theoretical studies that, because of the success of disciplining, extortion would largely disappear, we need no longer expect it in the real world. We challenge this view with our results.

What is new here is that contrary to this expectation disciplining does not discipline extortioners when players need to be competitively superior to gain an extra bonus. Because this kind of scenario can often be found in our society, our results revive the expectation for extortionate behaviour.

Appropriateness of statistical analysis

Yes

Reproducibility

I could not find exact experimental instructions and exact experimental procedure (e.g., time of day in which the experiment was conducted). So, as it stands now, the experiment cannot be replicated. However, I think this is pretty easy to fix.

Our fault -we thought we had attached the instructions to the Supplementary Information as is said in the method section, but unfortunately we forgot. This is fixed now, thank you for pointing this out.

Other comments:

The paper is generally well written, but I think it lacks some theoretical background: as I mentioned before, there is evidence that some people are inequity averse (Fehr & Schmidt, 1999). The existence of these people can certainly explain the results quite straightforwardly. I think that adding a discussion around would improve the paper.

Done, we have included a paragraph in the discussion making up for our neglect, citing not only Fehr & Schmidt 1999 but also Brosnan & de Waal 2014.

Reviewer #2 (Remarks to the Author):

[This review is in an attached PDF]

"Incentive for extra gain helps "extortion" to resist disciplining, without incentive "generous" wins" by Becks and Milinski explores the impact of asymmetry in incentives in an iterated Prisoner's Dilemma (IPD) game on the strategic behavior of human players. They show that such asymmetry results in lower rates of cooperation, more competitive behavior, lower payoffs during the game and a tendency among the most competitive players to employ "extortion" strategies. In contrast they show that players tend to default to "generous" strategies when such asymmetry is absent. This is a fascinating study whose results are of both theoretical and practical interest since they elucidate the relevance of extortion strategies to human behavior and the role of incentive structures in inducing such behaviors. I therefore believe that this study will be of considerable interest to the broad readership of Nature Communications and I support publication.

Thank you very much for your appreciation.

Major comments

The authors are careful to discuss their strategies as “potentially generous” and “potentially extortionate” and this is correct since they are projecting the player's actual behavior onto the space of ZD strategies. One striking aspect of their work is that, while Figure 2 clearly shows that extortion-like behavior is occurring in the two asymmetric conditions and generous-like behavior is occurring in the control condition, nonetheless the extortion-like behavior is still not strictly extortion in the originally defined sense of Press & Dyson 2012 and subsequent theoretical studies. This is not to dispute their findings but simply to point out that extortion and generosity appear as spectrum in their work and indeed there are even case particularly in T2, where low-levels of cooperation lead to the extortion-like strategy to score less than their partner. In This sense players may be thought of as permitting “regions” of generosity and regions of extortion in their interactions, with the out-of game pay-off shifting the balance of what players are willing to accept. I think this aspect of the empirical results is worth highlighting as it is interesting and should inform the way subsequent theoretical, computational and experimental studies are structured.

Thank you for adding your expert understanding to our interpretation, which in this way gains realism that will be appreciated by our readers. We have included some of your wording to the discussion section.

REVIEWERS' COMMENTS:

Reviewer #1 (Remarks to the Author):

Thanks for addressing my comments.

Reviewer #2 (Remarks to the Author):

The authors have revised the manuscript substantially in response to the comments of reviewers and I am happy to support publication in its current form.